# Training on Fake Labels: Mitigating Label Leakage in Split Learning via Secure Dimension Transformation

## Abstract

Two-party split learning has emerged as a popular paradigm for vertical federated learning. To preserve the privacy of the label owner, split learning utilizes a split model, which only requires the exchange of intermediate representations (IRs) based on the inputs and gradients for each IR between two parties during the learning process. However, split learning has recently been proven to survive label inference attacks. Though several defense methods could be adopted, they either have limited defensive performance or significantly negatively impact the original mission. In this paper, we propose a novel two-party split learning method to defend against existing label inference attacks while maintaining the high utility of the learned models. Specifically, we first craft a dimension transformation module, SecDT, which could achieve bidirectional mapping between original labels and increased $K$-class labels to mitigate label leakage from the directional perspective. Then, a gradient normalization algorithm is designed to remove the magnitude divergence of gradients from different classes. We propose a softmax-normalized Gaussian noise to mitigate privacy leakage and make our $K$ unknowable to adversaries. We conducted experiments on real-world datasets, including two binary-classification datasets (Avazu and Criteo) and three multi-classification datasets (MNIST, FashionMNIST, CIFAR-10); we also considered current attack schemes, including direction, norm, spectral, and model completion attacks. The detailed experiments demonstrate our proposed method's effectiveness and superiority over existing approaches. For instance, on the Avazu dataset, the attack AUC of evaluated four prominent attacks could be reduced by 0.4532±0.0127.

## 1 Introduction

Deep learning has been applied in many areas of people's daily lives. However, the paradigm of data-centralized deep learning has been continuously questioned since people's concerns about their privacy rose. For a typical scene, consider that an online shopping company A owns the clients' purchase records, while an online video website owner B has the clients' advertisement data. To learn how the advertisement on B will impact clients' purchase tendency on A. A and B need to learn a global model using A's data labels and B's features. However, due to privacy concerns and regulations such as GDPR for clients' data, A and B cannot directly share their data to train a global model Turina et al. (2021); Thapa et al. (2021); Zhang et al. (2023); Turina et al. (2020).

Split learning Yang et al. (2019); Gupta & Raskar (2018); Poirot et al. (2019); Vepakomma et al. (2018); Liu et al. (2024) is a solution to such a scenario, allowing the feature and label owners to train a machine learning model jointly. In split learning, the neural network's training process is split into the non-label and label parties Langer et al. (2020). At the beginning of split learning, the two parties apply Private Set Intersection (PSI) protocols Abuadbba et al. (2020) to find the intersection of their data examples and establish the alignment of data example IDs. During training, the non-label party will use a bottom model to obtain the intermediate layer of its data examples and send it to the label party. Then, the label party will complete the rest of the neural network training process. The label party will apply a top model to predict the non-label party's data examples. Next, in the backpropagation process, the label party computes the gradient from its loss function and sends the parameter gradient back to the non-label party. Thus,

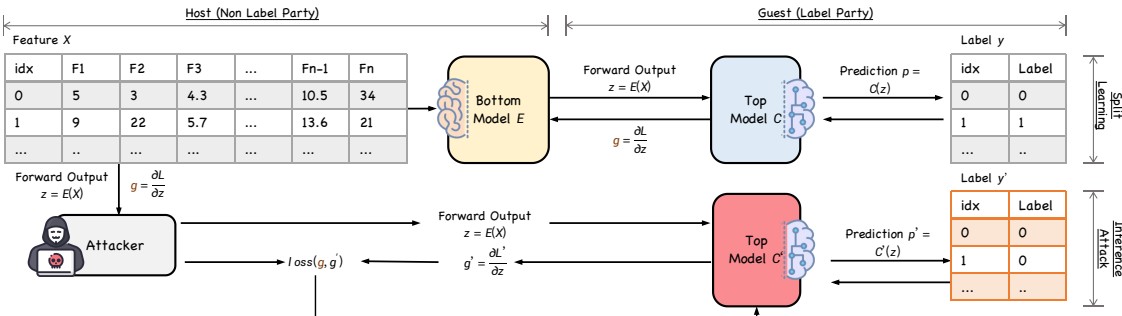

Figure 1: In the threat model, we assume the host to be the attacker who wants to get the private data label owned by the guest. This inference attack can be done through the inference attack from the backward gradient.

the non-label party can take the parameter gradient from the label party to train its bottom model. Split learning has been applied in many areas Pham et al. (2023); Abuadbba et al. (2020); Lin et al. (2024); Wu et al. (2023b); Matsubara et al. (2022); Samikwa et al. (2022); Wu et al. (2023a); Park et al. (2021).

Though the goal of split learning is to preserve privacy for both parties, the challenge of privacy leakage still exists within split learning. The non-label party can establish a gradient inversion attack Kariyappa & Qureshi (2021), a label leakage attack that allows an adversarial input owner to learn the label owner's private labels by exploiting the gradient information obtained during split learning. In the gradient inversion attack, the label leakage attack is treated as a supervised learning problem by developing a novel loss function using specific key properties of the dataset and model parameters instead of labels. The non-label party can derive the labels of data examples via a gradient inversion attack, thus harming the privacy of the label party. Split learning urgently requires a protection mechanism to guard the privacy of label parties. In this paper, we propose these contributions to achieve this goal:

- We craft a dimension transformation module that could achieve bidirectional mapping between original and increased $K$-class labels to mitigate label leakage from the directional perspective.
- A gradient normalization algorithm is designed to remove the magnitude's divergence of gradients w.r.t. samples from different classes.
- We propose one softmax-normalized Gaussian noise to mitigate privacy leakage and make our $K$ unknowable to adversaries.

## 2 Background and Related Work

### 2.1 Privacy Leakage in Split Learning

Though in split learning, the label party and non-label party will only share intermediate representation, privacy leakage threats still exist in this system Geiping et al. (2020); Jin et al. (2021); Wu et al. (2022); Jiang et al. (2022); Qi et al. (2022).

**Feature leakage**. First, the forward intermediate representations can leak the private feature information in the non-label party. Vepakomma et al. (2019) is the first to such feature leakage in split learning and provided a defense solution using distance correlation. By creating a novel loss function employing specific key properties of the dataset and model parameters, Kariyappa & Qureshi (2021) created a Gradient Inversion Attack (GIA), which converts the attack into a supervised learning problem.

**Label leakage**. Second, the backward intermediate representation can leak the private label information in the label party. Li et al. (2022) propose a norm-based method for leaking private labels in the conversion prediction problem. Their method is inspired by the high-class imbalance of the training dataset for the conversion prediction task. Due to this imbalance, the gradients' magnitude is more significant when the unusual class is encountered. So, an adversarial input owner can infer the private class labels by considering the gradients' norm. Instead, they investigate whether label information may be disclosed through backward

interaction from the label party to the non-label party. Fu et al. (2022) discover that a malicious participant can exploit the bottom model structure and the gradient update mechanism to gain the power to infer the privately owned labels. Worse still, by abusing the bottom model, he/she can infer labels beyond the training dataset. Based on their findings, they propose a set of novel label inference attacks. Sun et al. (2022) uses SVD to find correlations between embeddings and labels. Analyzing the mean and singular vectors assigns scores, clusters samples, and infers labels, bypassing privacy protections. Liu et al. (2021) first investigate the potential for recovering labels in the vertical federated learning context with HE-protected communication and then demonstrate how training a gradient inversion model can restore private labels. Additionally, by directly substituting encrypted communication messages, they demonstrate that label-replacement backdoor attacks may be carried out in black-boxed VFL (termed "gradient-replacement attack").

Our work SECDT focuses on defending label leakage threats.

## 2.2   Privacy Protection in Split Learning

Techniques to protect communication privacy in FL generally fall into three categories: 1) cryptographic methods such as secure multi-party computation Bonawitz et al. (2017); Zhang et al. (2020); Wan et al. (2024); Pereteanu et al. (2022); 2) system-based methods including trusted execution environments Subramanyan et al. (2017); and 3) perturbation methods that shuffle or modify the communicated messages Abadi et al. (2016). Defense against label leakage threats can be categorized into the 3rd category.

Previous researchers have proposed several defense schemes to defend against label leakage threats. Li et al. (2022) propose Marvell, which strategically determines the form of the noise perturbation by minimizing the label leakage of a worst-case adversary. The noise is purposefully designed to reduce the gap between the gradient norms of the two classes, which deters the attack. However, it requires a large amount of extra computation that slows down the speed of split learning. Abadi et al. (2016) proposed using differential privacy in deep learning to protect privacy. We utilize differential privacy in vertical federated learning on the guest side to safeguard the confidentiality of data labels. Wei et al. (2023) propose MixPro, an innovative defense mechanism against label leakage attacks in Vertical Federated Learning (VFL) that employs a two-step approach: Gradient Mixup and Gradient Projection. To prevent the non-label party from assuming the genuine label, Liu et al. (2021) also introduced "soft fake labels." To increase confusion, a confusional autoencoder (CoAE) is used to create a mapping that transforms one label into a soft label with a more significant likelihood for each alternative class.

## 3   Label Inference Attack

In this section, we first show the system model of split learning and threat model of label inference attacks, followed by five kinds of label inference attacks.

## 3.1   System Model

In two-party split learning, considering a training dataset $\{X_i, y_i\}_{i=1}^N$, the host (non-label party) has the features $\{X_i\}_{i=1}^N$ and the guest (label party) has the corresponding labels $\{y_i\}_{i=1}^N$ as shown in Figure 1. They jointly train a model based on the training dataset $\{X_i, y_i\}_{i=1}^N$ without leaking $\{X_i\}_{i=1}^N$ and $\{y_i\}_{i=1}^N$ to each other. Specifically, the host trains a bottom model $E$ while the guest trains a top model $C$. In each iteration, the host sends the forward embedding output $z = E(X)$ of the bottom model $E$ to the guest, then the guest will use $z$ as the input of the top model $C$ and send the backward gradient $g$ to the host. After receiving the gradient $g$, the host uses $g$ to update the bottom model code $E$. The host will not access the label $\{y_i\}_{i=1}^N$, while the guest will not access the data feature $\{X_i\}_{i=1}^N$, thus protecting the privacy of both the host and the guest.

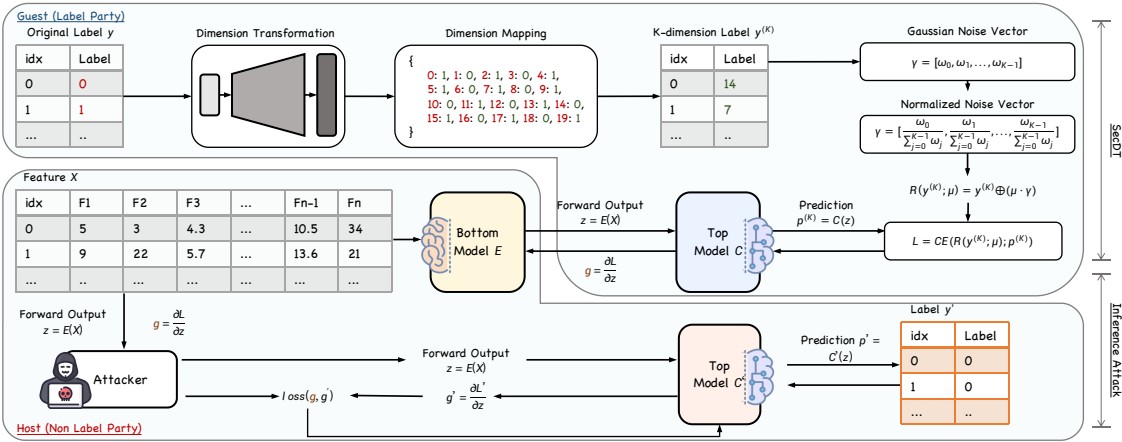

Figure 2: This figure outlines the SecDT algorithm's training process, which includes dimension transformation to expand the label space, gradient normalization to standardize gradient magnitudes, and noise-based randomization to introduce uncertainty into the label dimensions.

## 3.2 Potential Attacks

Then we consider the malicious host who wants to get the label held by the guest. We consider five types of attacks in our threat model:

**Direction Attack Li et al. (2022)** In this attack, for a given example, all same-class examples show positive cosine similarity, while opposite-class examples yield negative similarity. In class-imbalanced scenarios, if the non-label party knows the majority class, they can infer labels: a majority positive similarity implies a negative class, otherwise positive. This method leverages population knowledge, such as the rarity of certain diseases in prediction tasks.

**Norm Attack Li et al. (2022)** This attack leverages model training confidence, where the model is typically less certain about positive examples. Positive examples tend to yield higher gradient norms $||g||_2$, making the gradient norm $r_n(g) = ||g||_2$ a reliable predictor of the hidden label $y$. The resulting privacy loss is measured as norm leak AUC.

**Spectral Attack Sun et al. (2022)** This attack applies singular value decomposition (SVD) on intermediate embeddings to reveal label correlations. By calculating mean and top singular vectors, the attack clusters samples and infers labels, bypassing privacy-preserving techniques.

**Model Competition Attack Fu et al. (2022)** Here, an adversary uses a pre-trained bottom model, adding random inference layers and fine-tuning with a small labeled set to form a complete model. Using semi-supervised learning, they can predict labels for any sample, thus passively inferring labels without actively modifying the VFL setup.

## 4 The Proposed SecDT

### 4.1 Overview

Figure 2 shows the training workflow of our proposed SECDT, which comprises three parts, i.e., dimension transformation, gradient normalization, and randomization. In two-party split learning, there is a host (feature party) and a guest (label party), but only the guest has the training labels. However, the private label information could be inferred by the offensive host from the backward gradients $\{\mathbf{g}_i\}_{i=1}^N$. Generally speaking, existing label inference attacks against split learning are designed from two perspectives: direction and magnitude Li et al. (2022); Sun et al. (2022); Fu et al. (2022). Therefore, the dimension transformation is crafted to achieve bidirectional mapping between original and increased $K$-class labels, mitigating label

leakage from the directional perspective. Then, to remove the magnitude's divergence of gradients, we design a magnitude normalization algorithm to achieve this goal. Furthermore, we introduce a randomization module in which two random noises are proposed to mitigate privacy leakage and make our $K$ unknowable to adversaries. All the defenses are conducted by the guest, which is invisible to the host.

## 4.2 Dimension Transformation

It is intuitive that for the adversaries, the original classification tasks are more vulnerable to label inference attacks than the multi-class classification tasks, especially from the directional perspective.

**Dimension-increased transformation** Could we transform original labels to multi-class labels during model training? To achieve this goal, we craft the dimension transformation module. Transfer from binary to multi-class: Specifically, given dataset $D = \{X_i, y_i\}_{i=1}^N$ with labels $\{y_i\}_{i=1}^N$, we have

$$\text{One-hot}(y_i) = \hat{y}_i \in \text{One-hot}(\{0, \cdots, k-1\}), \tag{1}$$

where One-hot$(\cdot)$ converts numbers 0 to $k-1$ to one-hot encodings. For instance, if $k = 3$ (i.e., the task is a three-category task), we have One-hot$(0, 1, 2) = \{[1, 0, 0], [0, 1, 0], [0, 0, 1]\}$. Then, given labels $\{y_i\}_{i=1}^N$ and targeted (increased) dimension $K$, we define a mapping $M_{k,K}$ from $k$-dimension to $K$-dimension as

$$\mathcal{M}_{k,K}(y_i) = m_{K,y_i}, \tag{2}$$

where $m_{K,y_i}$ are elements that randomly selected from corresponding mapping pool $\rho_{K,y_i}$ for label $y_i$. To generate the required mapping pools $\rho_{K,y_i}$, with specific $K$, an ordered set of one-hot encodings is defined as

$$\Psi = \{\text{One-hot}(0), \cdots, \text{One-hot}(K-1)\}. \tag{3}$$

Then, we shuffle $\Psi$ randomly or according to specific rules to get

$$\Psi_s = \text{Shuffle}(\Psi) \tag{4}$$

and separate $\Psi_s$ into $k$ disjoint mapping pools given by

$$\rho_{K,0}, \cdots, \rho_{K,k-1}, \tag{5}$$

where $\bigcap_{y=0}^{k-1} \rho_{K,y} = \emptyset$ and $\bigcup_{y=0}^{k-1} \rho_{K,y} = \Psi_s$. There could be many rules for dividing $\Psi_s$ to $\rho_{K,y}$. For simplicity, in this work, unless otherwise mentioned, we divide $\Psi_s$ equally to $\rho_{K,0}, \cdots, \rho_{K,k-1}$. Indicating the length (number of elements) of $\rho_{K,y}$ by $\sigma_y = \frac{K}{k}$, for each $\rho_{K,y}$, we have

$$\rho_{K,y} = \{\tau_{y,0}, \cdots, \tau_{y,\sigma_y-1}\}. \tag{6}$$

Then, we have the dataset $D_K = \{X_i, \mathcal{M}_{k,K}(y_i)\}_{i=1}^N$ with increased the $K$-dimension labels to optimize the objective

$$\min_{E,C} \mathcal{L}(\theta, D_K) = \min_{E,C} \frac{1}{N} \sum_{i=1}^N L\left(C(E(X_i)), \mathcal{M}_{k,K}(y_i)\right), \tag{7}$$

where $L(\cdot, \cdot)$ is for computing the cross entropy. Because dimension transformation works before training of the model and the dimension-increased labels are fixed, optimizing the Equation 7 could also optimize the objective

$$\min_{E,C} \mathcal{L}(\theta, D) = \min_{E,C} \frac{1}{N} \sum_{i=1}^N L\left(E(C((X_i)), \hat{y}_i\right). \tag{8}$$

**Dimension-decreased transformation** The $K$-dimension prediction could not be directly used as the final result during inference time. Intuitively, with the $K$-dimension prediction $p^{(K)} = C(z) = C(E(X))$, we could derive original $k$-dimension inference result $p$ based on a maximum mapping function $\mathcal{MM}(\cdot)$ given by

$$p = \mathcal{MM}(p^{(K)}) = y, \; if \; p^{(K)} \in \rho_{K,y}. \tag{9}$$

However, as turning a $k$-classification task into a $K$-classification task will increase the difficulty of model training, the aforementioned function $\mathcal{MM}(\cdot)$ results in compromised model performance compared with the original $k$-classification task. We believe this is because the data whose labels belong to the same mapping pool $\rho_{K,y}$ essentially have similar characteristics. *We believe this is because the data samples used for inference are similar but not identical to the training samples, resulting in the $p^{(K)}$ to be linear combinations of one-hot encodings in $\rho_{K,y}$ for feature $X$.* To overcome this challenge, in our SECDT, we propose to realize performance-reserved dimension-decreased transformation based on a novel weighted mapping function $\mathcal{WM}(\cdot)$ given by

$$p = \mathcal{WM}(p^{(K)}) = \arg\max_{0 \leq y \leq k-1} \mathbf{w}_y \cdot p^{(K)},\tag{10}$$

where $P^{(K)}$ is the weight. Here, $\mathbf{w_y} \cdot p^{(K)}$ is the inner product of $\mathbf{w}_y$ and $\cdot p^{(K)}$, where $\mathbf{w}_y$ represents the result of element-wise addition ($\oplus$) of one-hot encodings in the mapping pool $\rho_{K,y}$ that

$$\mathbf{w_y} = \tau_{y,0} \oplus \tau_{y,1} \oplus \cdots \oplus \tau_{y,\sigma_y-1}.\tag{11}$$

With function $\mathcal{WM}(\cdot)$, we covert the linear combination of one-hot encodings in $\rho_{K,y}$ to the inner product as the confidence that $P^{(K)}$ belongs to label $y$. Our results show that the proposed weighted mapping significantly guarantees the effectiveness of the task model.

### 4.3 Gradient Normalization

Inspired by Cao et al. (2021) that normalizes gradients from suspicious clients in horizontal federated learning, in SECDT, we make the first attempt to normalize gradients in the cut layer of split learning, which could fully avoid label inference attacks conducted based on the gradients' magnitude. Cao et al. (2021) normalize gradients with a trust-worthy gradient locally computed by the server, but there is no single trust-worthy gradient in our SECDT. Hence, since all gradients in our SECDT are clean (i.e., have not been manipulated by adversaries), literally, each gradient could be used to normalize others. In our SECDT, the minimum, mean, and maximum $\ell_2$-norm among all gradients in the current mini-batch could be used to realize normalization. In this paper, unless otherwise mentioned, the mean $\ell_2$-norm is considered as the *standard norm* for normalization. Specifically, when the batch size is set to be $B$, during each iteration, $B$ gradients $\{g_b\}_{b=1}^B$ are computed by the guest. Then, we normalize these gradients as

$$\bar{g}_b = g_b \cdot \frac{\varphi}{\|g_b\|},\tag{12}$$

where $\|\cdot\|$ represents $\ell_2$-norm and $\varphi$ is the selected standard $\ell_2$-norm. For instance, considering the mean $\ell_2$-norm as the standard, we have

$$\varphi = \frac{1}{B}\sum_{b=1}^{B}\|g_b\|.\tag{13}$$

### 4.4 Noise-based Randomization

Considering potential adaptive attacks against our proposed SECDT, which may succeed after an attacker could infer our mapping pools $\rho_{K,y}$. Hence, we aim to keep our increased dimension $K$ confidential to adversaries, making mapping pools unknowable. In this work, we propose adding Softmax-normalized Gaussian noise (SGN) to make our increased dimension $K$ agnostic to adversaries. Due to the task independence of introduced noise, the noise could also mitigate privacy leakage of split learning to existing attacks. We assume that during SECDT's model training phase, each sample's label (target) is a $K$-dimension one-hot encode $\tau = [\zeta_1, \zeta_2, \cdots, \zeta_K]$. Then, we propose two noises as follows.

**Softmax-normalized Gaussian noise** For each sample's one-hot encoded label $\hat{y}$, the guest generates a noise vector

$$\gamma = [\omega_0, \omega_1, \cdots, \omega_{K-1}],\tag{14}$$

where $\omega$ follows standard Gaussian distribution, i.e., $\omega \sim \mathcal{N}(0,1)$. Moreover, to make this noise more controllable, the guest normalizes $\gamma$ based on the softmax function as

$$
\begin{aligned}
\bar{\gamma} =& \text{Softmax}(\gamma) \\
=& [\frac{e^{\omega_0}}{\sum_{j=0}^{K-1} e^{\omega_j}}, \frac{e^{\omega_1}}{\sum_{j=0}^{K-1} e^{\omega_j}}, \cdots, \frac{e^{\omega_{K-1}}}{\sum_{j=0}^{K-1} e^{\omega_j}}]
\end{aligned}
\tag{15}
$$

**Adding noise** With generated noise for each sample's label $\tau$, we add them into the initial label to obtain

$$
\tau = \tau \oplus (\mu \cdot \gamma),
\tag{16}
$$

where $\mu$ is used to determine the noise level.

## 5 Evaluation Setup

In this section, we first discuss the datasets (§5.1), model architecture, and environment (5.2) for evaluation. Then, we introduce the compared schemes (§5.3) in our evaluation.

### 5.1 Datasets

We selected three image datasets for the multiple classifications and two click prediction datasets for the binary classification:

- **Criteo** dat (2014b): Criteo is a CTR dataset provided by Criteo. The training set consists of some of Criteo's traffic over seven days. Each row corresponds to a display ad served by Criteo.
- **Avazu** dat (2014a): Avazu is one of the leading mobile advertising platforms globally. It consists of 10 days of labeled click-through data for training and one day of ads data for testing (yet without labels). Only the first ten days of labeled data are used for benchmarking.
- **MNIST** LeCun et al. (1998): MNIST is a dataset of 70,000 28x28 pixel grayscale images of handwritten digits, split into 60,000 for training and 10,000 for testing.
- **FashionMNIST** Xiao et al. (2017): FashionMNIST is a 10-category dataset of 70,000 fashion item images, serving as a variation of MNIST for image classification tasks in machine learning.
- **CIFAR-10** Krizhevsky (2009): CIFAR-10 is a dataset with 60,000 32x32 color images across ten object classes for benchmarking image classification models.

Furthermore, we used AUC as the metric for the binary classification tasks, while we used accuracy as the metric for the multiple classification tasks.

### 5.2 Experiment Envioronment

For the three multiple-image classification datasets, we applied a CNN for classification. We used a WideDeep neural network to evaluate the two binary click classification datasets. All experiments are performed on a workstation equipped with Intel(R) Xeon(R) CPU E5-2650 v4 @ 2.20GHz, 32GB RAM, and four NVIDIA RTX 3060 GPU cards. We use PyTorch to implement DNNs.

### 5.3 Evaluated Schemes

We evaluated five label inference attack defense schemes in our experiments:

- **No defense**. The pure classification task in split learning without any defense mechanisms.
- **Marvell**. Li et al. (2022) proposed a random perturbation technique, which strategically finds the structure of the noise perturbation by minimizing the amount of label leakage (measured through our quantification metric) of a worst-case adversary (called Marvell).
- **DP**. Abadi et al. (2016) proposed using differential privacy in deep learning to protect privacy. We utilize differential privacy in vertical federated learning on the guest side to protect the privacy of data labels.

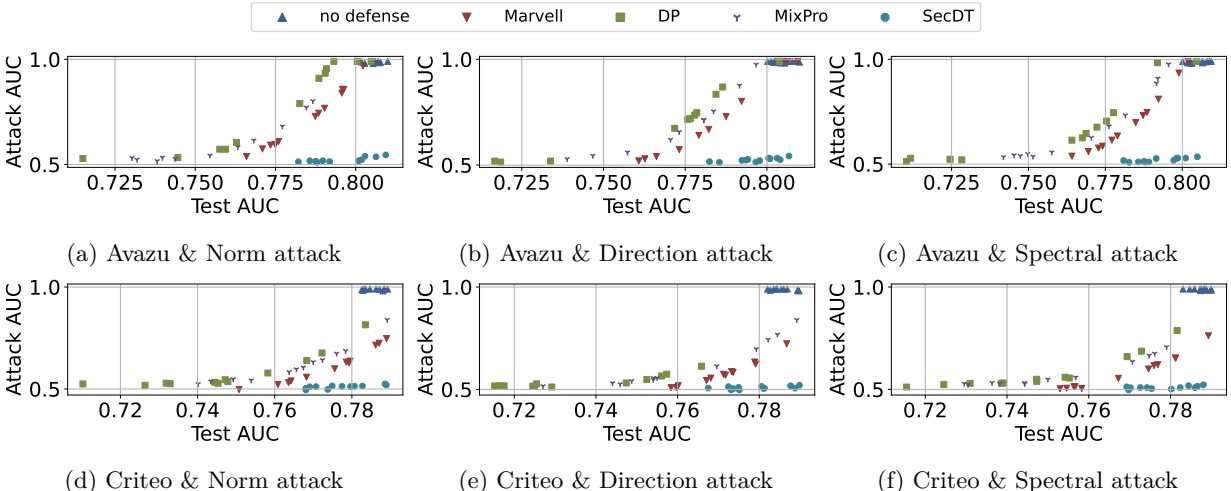

(a) Avazu & Norm attack    (b) Avazu & Direction attack    (c) Avazu & Spectral attack

(d) Criteo & Norm attack    (e) Criteo & Direction attack    (f) Criteo & Spectral attack

Figure 3: This figure presents the results of SecDT's effectiveness against label inference attacks compared to other defense schemes such as Marvell, DP, and MixPro. It shows SecDT's superior performance in reducing Attack AUC without compromising Test AUC.

- **MixPro**. MixPro Wei et al. (2023) is an innovative defense mechanism against label leakage attacks in Vertical Federated Learning (VFL) that employs a two-step approach: Gradient Mixup and Gradient Projection.
- **SecDT**. We proposed a dimension transformation method called SECDT, which transforms the classification task into a fake label classification task on the guest side.

## 6 Evaluation Results

**Comparison with Other Schemes**

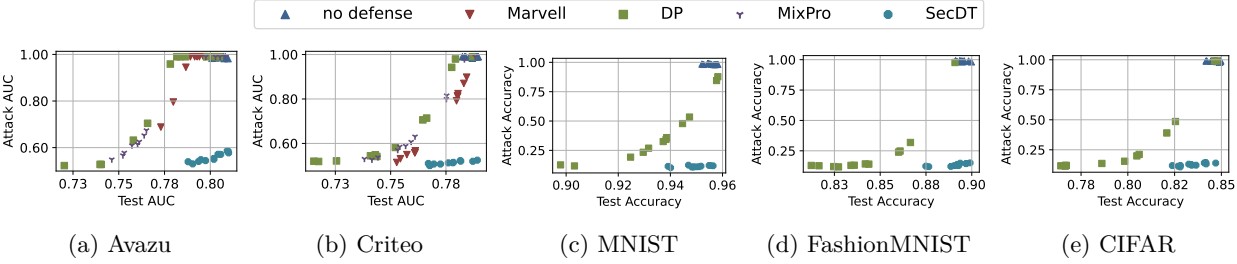

(a) Avazu    (b) Criteo    (c) MNIST    (d) FashionMNIST    (e) CIFAR

Figure 4: This figure evaluates SECDT's performance against model completion attacks, where an adversary obtains a bottom model and fine-tunes it with auxiliary labeled data. SECDT demonstrates robustness against such attacks, with less impact on performance than other defense schemes.

The evaluation results, depicted in Figure 3 and Figure 4, provide a comparative analysis of SECDT against other state-of-the-art defense mechanisms. The Test AUC (Area Under the Curve) represents the model's performance on legitimate tasks, while the Attack AUC represents the model's utility for adversarial tasks. The balance between these two metrics is crucial in determining the effectiveness of a defense mechanism.

The results indicate that SECDT outperforms other defense schemes such as Marvell, DP (Differential Privacy), and MixPro in defending against label inference attacks without significantly compromising the model's accuracy. Specifically, SECDT demonstrated lower Attack AUC scores across norm, direction, and spectral attacks on the Avazu and Criteo datasets, signifying its superior ability to resist adversarial attacks.

Model completion attacks were also evaluated, involving an adversary obtaining a bottom model and fine-tuning it with auxiliary labeled data. The results show that SECDT's performance is less affected by this

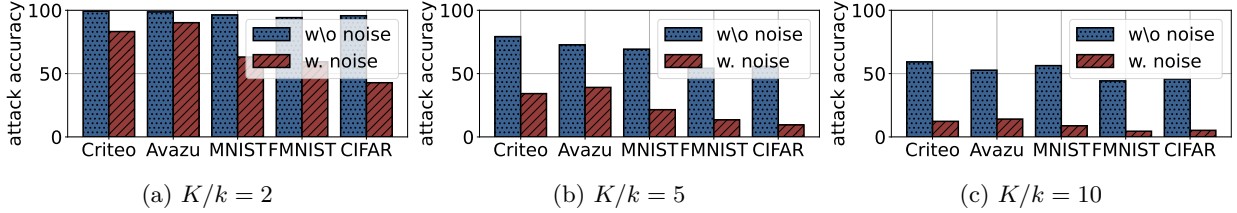

Figure 5: The attack accuracy of a potential attacker using K-means based method for inferring the value of $K$

attack than other defense schemes. This indicates that SECDT's multi-faceted approach provides a more robust defense against various adversarial strategies.

**Inference Attack towards $K$ in SecDT**

When an attacker knows our SECDT is used to learn the global model, it can adapt its attacks to SECDT. The attacker can guess the increased dimension $K$ in SECDT. Since the mapping from 2-dimension to $K$-dimension $\mathcal{M}_{2,K}$ will reveal some clustering features. We evaluate whether the attacker can guess the real value of $K$ in SECDT. For each data sample's backward gradient $g$, we iteratively set the dimension size from 2 to $2K$. Then, we use the K-means clustering algorithm to cluster the gradient to the set dimension number of clusters. After that, the calinski harabasz score is used to evaluate the clustering result. The set dimension with the highest will be elected as the guessed dimension. We set the value of $K/k$ to be 2, 5, and 10 in our experiment.

We evaluate SECDT w/ and w/o noise in Figure 5, and the result shows that the attacker can easily guess the dimension $K$ of SECDT since the frequency it guesses right is very high. Furthermore, we evaluate SECDT with our proposed SGN and find out that the frequency that the attacker guesses the right dimension $K$ becomes very low. Thus, the result indicates that our proposed noise can ease the proposed adaptive attack toward SECDT. **Time Cost**

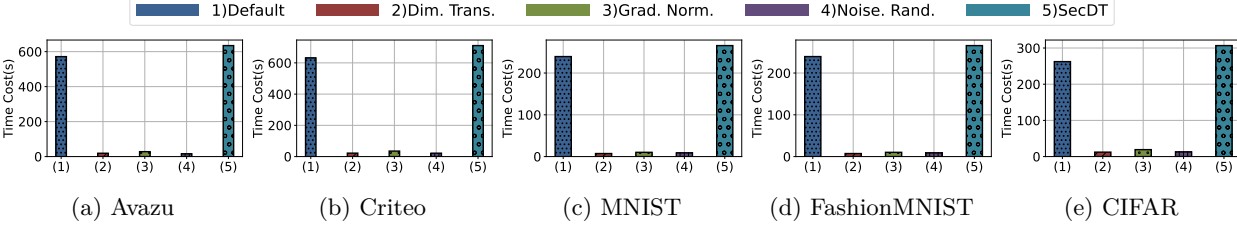

Figure 6: This figure evaluates SECDT's time cost on five evaluated datasets for different processes: **1) Default**: The default split learning without any defense. **2) Dim. Trans.**: The dimension transformation process of SECDT. **3) Grad. Norm.**: The gradient normalization process of SECDT. **4) Noise. Rand.**: The noise-based randomization process of SECDT. **5) SecDT**: The whole process of SECDT (including 1)+2)+3)+4)). The evaluation results show that the time cost of SECDT is not greatly increased compared with default split learning.

To comprehensively assess SECDT, we measured the time costs of its key processes, including dimension transformation, gradient normalization, and noise-based randomization. Experiments on datasets like Avazu, Criteo, MNIST, FashionMNIST, and CIFAR demonstrated a modest 11% increase in time cost compared to baseline split learning without defenses (Figure 6).

This overhead stems from secure operations such as higher-dimensional label mapping for privacy, gradient l2-norm computation for normalization, and noise injection for added randomness. Despite the increase, the trade-off is justified by the enhanced security and privacy, making SECDT a scalable and practical solution for split-learning applications.

| atk. type | norm attack | direction attack | spectral attack | mod. comp. |
|---|---|---|---|---|
| Criteo | | | | |
| w/o decrease | 49.91/57.89 | 52.06/61.37 | 52.06/60.64 | 59.60/61.19 |
| decrease | 52.34/76.95 | 55.57/78.10 | 55.17/77.48 | 62.87/76.55 |
| Avazu | | | | |
| w/o decrease | 49.78/59.01 | 52.36/58.8 | 51.99/58.2 | 59.57/59.59 |
| decrease | 52.52/76.94 | 54.92/74.77 | 55.23/76.32 | 63.42/75.42 |
| MNIST | | | | |
| w/o decrease | -/- | -/- | -/- | 16.06/72.43 |
| decrease | -/- | -/- | -/- | 16.94/97.61 |
| FashionMNIST | | | | |
| w/o decrease | -/- | -/- | -/- | 13.73/73.51 |
| decrease | -/- | -/- | -/- | 14.69/90.56 |
| CIFAR | | | | |
| w/o decrease | -/- | -/- | -/- | 12.54/66.5 |
| decrease | -/- | -/- | -/- | 13.3/84.80 |

Table 1: We compared SECDT without dimension-decreased transformation and SECDT with dimension-decreased transformation. This table recorded the attack and test utility as "atk uti./test uti.". The results indicated that dimension-decreased transformation can maintain the accuracy performance of SECDT while not comprising SECDT's ability to defend various label inference attacks.

## 7 Ablation Study

**Impact of Expanded Dimension Size**

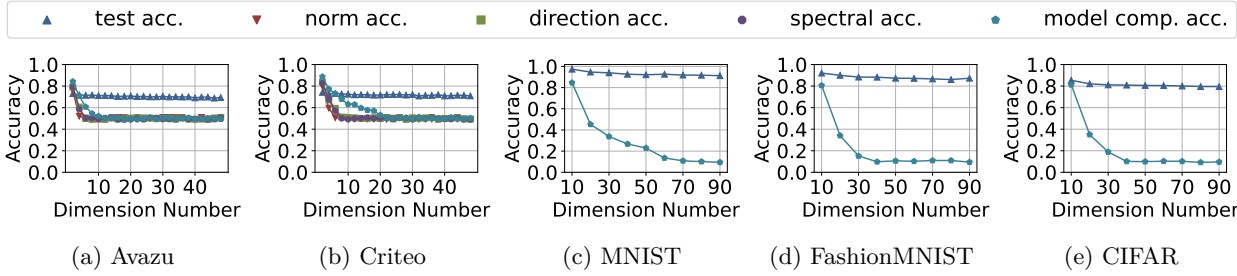

(a) Avazu     (b) Criteo     (c) MNIST     (d) FashionMNIST     (e) CIFAR

Figure 7: Normalization plays a vital role in defending against norm and model completion attacks.

We first evaluate the impact of expanded dimension size on the performance of split learning. We applied all four attack schemes for the Avazu and Criteo datasets, while for MNIST, Fashion-MNIST, and CIFAR-10 datasets, we only applied the model competition attacks.

As the results shown in Figure 7, the increasing dimension size does not affect the performance of the test utility. While the attack utility is strongly affected as the dimension size increases. For experiments on Avazu and Criteo, the norm, direction, and spectral attack utilization decrease soon and converge to the lower boundary as the dimension size reaches 8 (four times the original dimension size 2). The model completion attack stood steady until the dimension size reached 20 (10 times the original dimension size). This indicated that model completion attack requires SECDT to expand to a stronger dimension size to defend.

**Impact of Dimension Decrease**

Furthermore, we compared SECDT without dimension-decreased transformation and SECDT with dimension-decreased transformation. Dimension decrease is vital in keeping the accuracy of split learning when applying SECDT.

The dimension decrease transformation is a critical component of the SECDT framework that facilitates the conversion of the increased K-class labels back to the original binary or k-class labels. This process is

| atk. type | norm attack | direction attack | spectral attack | mod. comp. |
|---|---|---|---|---|
| Criteo | | | | |
| w/o random. | 57.31/77.78 | 65.66/76.59 | 65.50/77.64 | 70.47/76.84 |
| random. | 57.32/74.87 | 53.37/76.07 | 53.77/75.90 | 63.28/75.80 |
| Avazu | | | | |
| w/o random. | 57.84/74.83 | 66.42/76.36 | 65.58/76.17 | 71.61/75.22 |
| random. | 57.77/73.72 | 53.22/73.85 | 53.27/75.09 | 62.76/74.85 |
| MNIST | | | | |
| w/o random. | -/- | -/- | -/- | 28.26/96.22 |
| random. | -/- | -/- | -/- | 17.05/94.31 |
| FashionMNIST | | | | |
| w/o random. | -/- | -/- | -/- | 24.99/93.24 |
| random. | -/- | -/- | -/- | 14.41/90.21 |
| CIFAR | | | | |
| w/o random. | -/- | -/- | -/- | 22.37/85.62 |
| random. | -/- | -/- | -/- | 13.26/81.73 |

Table 2: We compared SECDT without normalization and SECDT with normalization. This table recorded the attack and test utility as "atk uti./test uti.". The results indicated that normalization can maintain the accuracy of SECDT while not comprising SECDT's ability to defend various label inference attacks. Normalization is vital in defending against norm and model completion attacks.

essential for the practical application of the model, as it ensures that the output is usable for the end-user. As detailed in §4.2, SECDT employs a weighted mapping function to achieve this transformation, designed to preserve the model's performance while safeguarding against label inference attacks.

Our experiments were conducted on diverse datasets, including Avazu, Criteo, MNIST, FashionMNIST, and CIFAR-10, to evaluate the impact of the dimension decrease transformation. These datasets span various domains, comprehensively assessing the technique's effectiveness.

The experimental results, as depicted in Table 1 of the original document, offer profound insights into the significance of the dimension decrease transformation in SECDT. When SECDT was applied without incorporating the dimension decrease, a notable decline in test accuracy was observed across all evaluated datasets. This decline underscores the importance of the transformation in maintaining the model's predictive power. Conversely, introducing the dimension decreased transformation and led to a substantial improvement in test accuracy. This enhancement was consistent across different datasets, demonstrating the technique's robustness. The results indicate that the weighted mapping function effectively navigates the increased dimensionality introduced during the training phase and accurately translates it back into the original label space. Furthermore, the results highlight that the dimension decrease transformation does not compromise SECDT's ability to defend against label inference attacks. The attack utility remained consistently low, even when the transformation was applied, validating the technique's efficacy as a privacy-preserving measure.

The dimension decrease transformation's impact was also assessed against various attack vectors, including norm attacks, direction attacks, spectral attacks, and model completion attacks. The results demonstrated that the transformation did not adversely affect SECDT's resilience against these attacks. This finding is particularly significant, as it suggests that the dimension decrease can be universally applied to enhance the security of split learning models without diminishing their defense capabilities.

An additional observation from the experiments is the interaction between the noise level and the effectiveness of the dimension decrease transformation. While a moderate noise level did not significantly impact test utility, higher noise levels did lead to a decline in accuracy. This observation suggests that the dimension decrease transformation can counteract the adverse effects of noise to a certain extent, providing an additional layer of protection for the model.

**Impact of Normalization**

We compared SECDT without normalization and SECDT with normalization. Normalization is vital in defending against norm attacks when applying SECDT.

SECDT's approach to gradient normalization, as detailed in §4.3, involves using the mean $l2$-norm of the gradients within a mini-batch as a standard for normalization. This method is particularly effective against attacks that rely on the magnitude of gradients to infer sensitive information. By enforcing a standard norm, SECDT diminishes the adversaries' ability to discern differences in gradient magnitudes that could indicate the presence of certain classes within the dataset.

The use of gradient normalization in SECDT introduces a trade-off between utility and privacy. While normalization can reduce the model's utility for adversaries, it may also affect the model's ability to learn complex patterns if not implemented carefully. However, our experiments, as shown in Table 2, demonstrate that SECDT successfully strikes a balance, maintaining high test accuracy while significantly reducing the attack utility.

**Impact of Noise**

Finally, we evaluated the impact of noise scale on split learning performance using four attack schemes for Avazu and Criteo datasets, and model competition attacks for MNIST, FashionMNIST, and CIFAR-10.

Noise-based randomization, a key feature of SECDT, obfuscates label mappings and prevents adversaries from identifying the true label dimensionality (§4.4). This complements dimension transformation and gradient normalization to strengthen defenses against label inference attacks.

Experiments tested varying noise levels on real-world datasets, assessing their effect on legitimate tasks and adversarial resilience. As shown in Figure 8, noise integration preserved model performance up to a threshold, with test utility declining only beyond a noise level of 0.5, marking a critical point where privacy comes at the cost of accuracy.

Conversely, noise significantly reduced attack utility, particularly for norm, direction, and spectral attacks, which were neutralized at noise levels above 0.2. However, model completion attacks remained resilient up to a noise level of 0.4, highlighting the need for layered defenses like dimension transformation and gradient normalization to bolster security further.

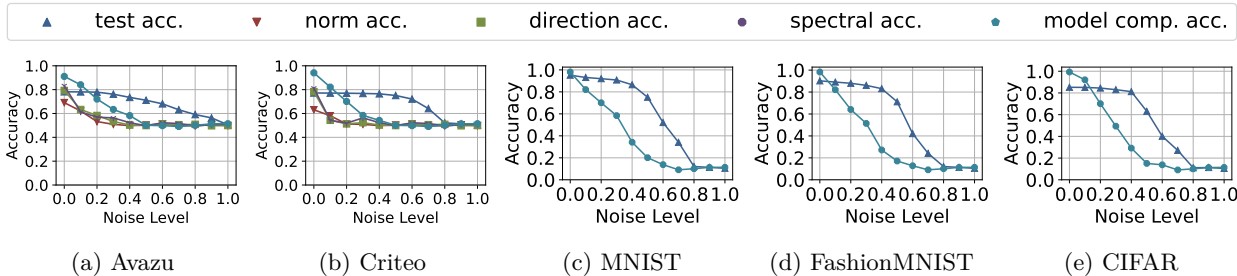

Figure 8: The evaluation results of noise's impact on SECDT's performance.

## 8    Conclusion

In this study, we present a novel two-party split learning strategy, SECDT, to counter label inference attacks that have already been made. To reduce label leakage from a directional perspective, we first create a dimension transformation module that could achieve bidirectional mapping between binary and enhanced $K$-class labels. Then, a gradient normalization algorithm is created to eliminate the amplitude of gradients from various classes diverging. We also provide two random noises to prevent privacy leaks and prevent attackers from knowing our $K$. Studies on two real-world datasets show how successful and superior our suggested solution is to other ways.

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

# A   Algorithm Description of SecDT

---

**Algorithm 1** Training on Fake Labels: SecDT Algorithm

---

1: **Input:** Dataset $D = \{X_i, y_i\}_{i=1}^N$, Desired dimension $K$, Noise level $\mu$
2: **procedure** DIMENSION-INCREASED TRANSFORMATION$(D, K)$
3:     $\hat{y}_i \leftarrow$ One-hot$(y_i)$                                                                   ▷ Convert labels to one-hot encoding
4:     $\Psi \leftarrow \{$One-hot$(0), ...,$ One-hot$(K - 1)\}$
5:     $\Psi_s \leftarrow$ Shuffle$(\Psi)$
6:     $\rho_{K,y} \leftarrow$ Partition$(\Psi_s, y)$                                                                   ▷ Create mapping pools
7:     $M_{k,K}(y_i) \leftarrow$ Map$(\hat{y}_i, \rho_{K,y_i})$                                                    ▷ Map to increased K-dimension
8:     $D_K \leftarrow \{X_i, M_{k,K}(y_i)\}_{i=1}^N$                                        ▷ Updated dataset with increased dimension labels
9: **end procedure**
10: **procedure** DIMENSION-DECREASED TRANSFORMATION$(p(K), \rho_{K,y})$
11:     $p \leftarrow$ WM$(p(K), \rho_{K,y})$                                               ▷ Weighted mapping function to original dimension
12: **end procedure**
13: **procedure** GRADIENT NORMALIZATION$(\{g_b\}_{b=1}^B)$
14:     $\phi \leftarrow \frac{1}{B} \sum_{b=1}^B \|g_b\|_2$                                                    ▷ Calculate mean l2-norm of gradients
15:     $g_b' \leftarrow \frac{g_b}{\|g_b\|_2} \cdot \phi$                                                                   ▷ Normalize gradients
16: **end procedure**
17: **procedure** NOISE-BASED RANDOMIZATION$(\tau, \mu)$
18:     $\gamma \sim \mathcal{N}(0, I_K)$                                                                       ▷ Generate Gaussian noise
19:     $\bar{\gamma} \leftarrow$ Softmax$(\gamma)$                                                           ▷ Normalize noise with Softmax
20:     $\tau' \leftarrow \tau \oplus (\mu \cdot \bar{\gamma})$                                                                   ▷ Add noise to labels
21: **end procedure**
22: **procedure** SECDT TRAINING$(D, K, \mu)$
23:     $D_K \leftarrow$ Dimension-Increased Transformation$(D, K)$
24:     **for** each mini-batch $\{g_b\}_{b=1}^B$ **do**
25:         $\{g_b'\} \leftarrow$ Gradient Normalization$(\{g_b\}_{b=1}^B)$
26:     **end for**
27:     **for** each sample's label $\tau$ **do**
28:         $\tau' \leftarrow$ Noise-Based Randomization$(\tau, \mu)$
29:     **end for**
30:     **Train** split learning models $E$ and $C$ using $D_K$
31:     **Output:** Models $E$ and $C$
32: **end procedure**

---

# B   Proof of Convergence

First, we made the following assumptions:

1. The loss function $L(\theta)$ is Lipschitz continuous, that is, there exists a constant $L > 0$ such that for any $\theta_1$ and $\theta_2$, we have $|L(\theta_1) - L(\theta_2)| \leq L\|\theta_1 - \theta_2\|$

2. The gradient $\nabla L(\theta)$ is bounded, that is, there exists a constant $G > 0$ such that for any $\theta$, we have $\|\nabla L(\theta)\| \leq G$

3. The learning rate $\eta_t$ satisfies the standard learning rate condition, that is, $\sum_{t=1}^\infty \eta_t = \infty$ and $\sum_{t=1}^\infty \eta_t^2 < \infty$

Next, we will analyze the impact of each step of SecDT on the loss function step by step and finally prove its convergence.

**Dimension conversion.** During the dimension conversion process, we convert the original k-dimensional label $y_i$ into a K-dimensional label $M_{k,K}(y_i)$. Assume that the original loss function is $L(\theta; X, y)$ and the converted loss function is $L'(\theta; X, M_{k,K}(y))$. We first analyze the change in the loss function.

Since the dimension conversion is only an expansion of the label space and does not affect the expressiveness of the feature space and the model, there is a fixed relationship between the original loss function and the converted loss function for a fixed model parameter $\theta$. Specifically, assuming that $\hat{y}$ is the predicted value of the original label and $\hat{y}'$ is the predicted value of the label after dimension conversion, we have:

$$L(\theta; X, y) = L'(\theta; X, M_{k,K}(y)) \tag{17}$$

**Noise Randomization.** In the noise randomization step, we add noise $\gamma$ to the label to obtain a new label $\tau' = \tau \oplus (\mu \cdot \bar{\gamma})$. The loss function after adding noise is $L''(\theta; X, \tau')$. We need to analyze the impact of noise on the loss function.

Since the noise $\gamma$ follows a standard Gaussian distribution and is normalized by the Softmax function, its amplitude is controllable. Assume the intensity of the noise is $\mu$, then for any $\theta$, we have:

$$L'(\theta; X, M_{k,K}(y)) \leq L''(\theta; X, \tau') + \mu \cdot \|\gamma\| \tag{18}$$

According to Gaussian noise's properties, the noise's effect $\gamma$ can be regarded as a constant term.

**Gradient Normalization.** In the gradient normalization step, we normalize each gradient. Assume the normalized gradient is $\bar{g}_b$, then:

$$\bar{g}_b = g_b \cdot \frac{\phi}{\|g_b\|} \tag{19}$$

Where $\phi$ is the mean $l_2$-norm of the gradient.

The purpose of gradient normalization is to balance the gradient amplitudes of different categories, thereby reducing the impact of the gradient amplitude on the loss function. Since the normalization operation of the gradient is linear, there is a fixed proportional relationship between the normalized gradient and the original gradient. That is, for any $\theta$ and gradient $\nabla L(\theta)$, there is:

$$\nabla L(\theta) \approx \bar{g}_b \tag{20}$$

**Convergence Proof.** We consider the entire training process of SecDT. Assume $\theta_t$ is the model parameter of the $t$th iteration, $\eta_t$ is the learning rate, and $\nabla L(\theta_t)$ is the current gradient, then the update rule is:

$$\theta_{t+1} = \theta_t - \eta_t \bar{g}_b \tag{21}$$

By the Lipschitz continuity assumption, for any $\theta_t$ and $\theta_{t+1}$, we have:

$$L(\theta_{t+1}) \leq L(\theta_t) - \eta_t \nabla L(\theta_t) \cdot \bar{g}_b + \frac{L}{2}\eta_t^2 \|\bar{g}_b\|^2 \tag{22}$$

Since $\bar{g}_b$ is the normalized gradient, whose norm is bounded, so there exists a constant $C > 0$ such that $\|\bar{g}_b\| \leq C$. Therefore, the above formula can be rewritten as:

$$L(\theta_{t+1}) \leq L(\theta_t) - \eta_t \|\nabla L(\theta_t)\|^2 + \frac{L}{2}\eta_t^2 C^2 \tag{23}$$

Summing the above formula and using the learning rate condition, we get:

$$\sum_{t=1}^{T} \eta_t \|\nabla L(\theta_t)\|^2 \leq L(\theta_1) - L(\theta_{T+1}) + \frac{L}{2}C^2 \sum_{t=1}^{T} \eta_t^2 \tag{24}$$

Since $\sum_{t=1}^{\infty} \eta_t^2 < \infty$, the sum on the right is bounded, let its upper bound be $B$, then:

$$\sum_{t=1}^{T} \eta_t \|\nabla L(\theta_t)\|^2 \leq L(\theta_1) - L(\theta_{T+1}) + B \tag{25}$$

Since $\sum_{t=1}^{\infty} \eta_t = \infty$, in order to ensure the left-hand side converges, $\|\nabla L(\theta_t)\|$ must approach zero. This means that the loss function $L(\theta_t)$ will converge to a bounded range.

In summary, we have proved that during the training of SecDT, the loss function $L(\theta)$ will converge to a bounded range, that is, there is a constant $M > 0$ such that:

$$\lim_{t \to \infty} L(\theta_t) \leq M \tag{26}$$

In order to determine the specific upper bound $M$, we need to analyze the convergence behavior of the above inequality term by term in detail. When $T \to \infty$, the sum of $\sum_{t=1}^{T} \eta_t \|\nabla L(\theta_t)\|^2$ will tend to $L(\theta_1) + B$.

Assume that the sum of $\|\nabla L(\theta_t)\|^2$ approaches a certain limit value $S$, then:

$$S = \sum_{t=1}^{\infty} \eta_t \|\nabla L(\theta_t)\|^2 \tag{27}$$

We can express the final upper bound $M$ as:

$$M = L(\theta_1) + B - S \tag{28}$$

Since $S$ represents the infinite sum of the squared sum of gradients, and according to $\sum_{t=1}^{\infty} \eta_t = \infty$ and $\sum_{t=1}^{\infty} \eta_t^2 < \infty$, we can further determine the specific value through the learning rate and gradient convergence behavior in the specific gradient descent algorithm.

**Conclusion** Based on the above analysis, we can determine the specific value of the constant $M$ as:

$$M = L(\theta_1) + \frac{L}{2} C^2 \sum_{t=1}^{\infty} \eta_t^2 - \sum_{t=1}^{\infty} \eta_t \|\nabla L(\theta_t)\|^2 \tag{29}$$

Where:

- $L(\theta_1)$ is the initial loss value

- $\frac{L}{2} C^2 \sum_{t=1}^{\infty} \eta_t^2$ is the upper bound of the noise effect

- $\sum_{t=1}^{\infty} \eta_t \|\nabla L(\theta_t)\|^2$ is the sum of the loss reduction during the gradient descent process

