# OpenReview forum: "Training on Fake Labels: Mitigating Label Leakage in Split Learning via Secure Dimension Transformation"
_TMLR — Rejected by TMLR_

### Review · Reviewer_sm1R · 2025-03-24

**Summary Of Contributions:**

Two-party split learning allows the feature and label owners to train a machine learning model jointly. It is typically used as a popular paradigm for vertical federated learning. To preserve the privacy of the label owner, split learning utilizes a split model, which only requires the exchange of intermediate representations (IRs) based on the inputs and gradients for each IR between two parties during the learning process rather than private raw inputs. Privacy preservation comes from not directly accessing the other party's private input. However, split learning has recently been proven to be vulnerable to label inference attacks.

To safeguard the privacy of label parties, this paper proposes three contributions: 1) crafting a dimension transformation module to increase the original classification to K-class labels to mitigate label leakage from the directional perspective; 2) A gradient normalization algorithm removing the divergence of gradients w.r.t. samples from different classes; 3) softmax-normalized Gaussian noise mechanism to mitigate privacy leakage and make our K unknow-able to adversaries.

**Audience:**

Yes

**Broader Impact Concerns:**

N.A.

**Claims And Evidence:**

No

**Requested Changes:**

Thanks for submitting your research to TMLR. According to the comments mentioned above, the requested changes are expected as,

- Add intuition behind the proposed method and privacy level in the front of section 4. Clarify the explicit privacy guarantee for each method.
- Add experimental comparisons of each proposed method in Section 4 with related works and baseline configurations of different baselines.
- Give a related work section to explicitly state which research direction our method follows. Instead of broadly discussing various domains (e.g., feature leakage, label leakage, cryptographic approaches, and trusted execution environments), this paper explicitly improves the gradient normalization or noise mechanism to highlight the explicitly proposed contribution instead of ambiguities.
- The missing references and comparisons in Section 4 are expected to be added.
- To make privacy guarantees more concrete, this work is expected to provide a quantitative evaluation of privacy leakage and complete proofs.

**Strengths And Weaknesses:**

Strengths
+ This paper explores an interesting and meaningful research direction, which involves privacy concerns of split learning. This topic is important.
+ The authors conducted experiments over five datasets, which are widely.

Weakness
- The paper did not give an intuition of the proposed methods to show why the proposed method is good. Simultaneously, it also remains unclear what level of privacy the proposed ideas provide.
- The experimental comparison and configurations are unclear. It is a little abnormal to compare the proposed work (not DP-based) with a DP work. Notably, the configurations of baseline works are unclear. For example, the choices of DP parameters are not mentioned. It is questioning how to make the comparison fair and aligned.
- Which research line followed by the authors is not clear. For example, related works generally discussed an extensive domain of works such as feature leakage, label leakage, secure multi-party computation from cryptography, and trusted execution environments. It is confusing which exploit research line the authors aim to improve.
- Section 4 lacks important references to dimension transformation and noise-based Randomization. It is unclear what is the difference between the proposed methods and the previous similar works. Concerning it, it is difficult to judge the real contribution of the proposed methods. To be specific, the discussion of related gradient normalization or dimension transformation is missing. The explicit comparisons in both related works and experimental evaluations are also missing.
- Although the paper presents the convergence proof, rigorous proofs of privacy guarantee were not explored or provided. It is essential part of a paper with the main contribution of proposing new privacy preservation. Particularly, given my understanding, the first contribution is something related to "anonymity", which increases the number of classes to mitigate the identification; while the other two are "removing memberships but keeping statistical attributes". The explicit privacy definition and the level of protection should be provided in a quantitative manner.

---

### Review · Reviewer_9JgL · 2025-03-27

**Summary Of Contributions:**

The paper introduces SecDT, a novel defense mechanism for two-party split learning aimed at mitigating label inference attacks while preserving model utility. The proposed method combines three key components:

Dimension Transformation: Expands the label space from k to K dimensions to obscure directional leakage.

Gradient Normalization: Standardizes gradient magnitudes to prevent magnitude-based attacks.

Noise-based Randomization: Adds softmax-normalized Gaussian noise to labels to further obscure the true label dimensionality.

The authors evaluate SecDT on binary (Avazu, Criteo) and multi-class (MNIST, FashionMNIST, CIFAR-10) datasets, demonstrating its effectiveness against four types of label inference attacks (direction, norm, spectral, and model completion). Results show significant reductions in attack success rates (e.g., attack AUC reduced by 0.4532 on Avazu) with minimal impact on model accuracy.

**Audience:**

Yes

**Broader Impact Concerns:**

I do not find any broader impact concerns

**Claims And Evidence:**

Yes

**Requested Changes:**

1.The combination of three techniques may complicate implementation and tuning in real-world scenarios.

2.The method's performance degrades when noise levels exceed 0.5, suggesting a trade-off between privacy and utility.

3. While effective against known attacks, the defense might not generalize to future, more sophisticated attacks.

4. Although modest, the 11% time cost increase could be prohibitive for resource-constrained applications.

5.The evaluation is limited to benchmark datasets; real-world deployment challenges (e.g., non-IID data) are not addressed.

**Strengths And Weaknesses:**

Comprehensive Defense: SecDT addresses multiple attack vectors (direction, magnitude, and adaptive attacks) through a multi-layered approach.

Theoretical and Empirical Validation: The paper provides a convergence proof for SecDT and extensive experiments across diverse datasets.

Practical Utility: The method maintains high model accuracy while defending against attacks, with only an 11% increase in computational overhead.

Innovative Techniques: The bidirectional dimension transformation and weighted mapping function are novel contributions to split learning privacy.

Adaptability: The noise-based randomization component makes the defense robust against adaptive attacks targeting the expanded dimension K.

---

### Review · Reviewer_YGgE · 2025-04-20

**Summary Of Contributions:**

This paper introduces SecDT, a novel privacy-preserving framework for two-party split learning that defends against label inference attacks without compromising model utility. It achieves this through a combination of label dimension transformation, gradient normalization, and noise-based randomization. By transforming labels into a higher-dimensional space and masking gradient patterns, SecDT effectively obscures sensitive label information from adversaries. Extensive experiments on five datasets demonstrate SecDT’s superior resilience against four types of label inference attacks while maintaining strong predictive performance and incurring minimal computational overhead, offering a practical and robust solution for privacy in vertical federated learning.

**Audience:**

Yes

**Claims And Evidence:**

Yes

**Requested Changes:**

- The paper would benefit from a dedicated subsection or extended discussion on how SecDT can be integrated into practical Vertical Federated Learning (VFL) or Split Learning (SL) systems. This should cover protocol-level adaptations, potential communication overheads, compatibility with existing privacy-preserving mechanisms (e.g., PSI, HE), and deployment strategies in cloud or cross-organization environments.
- Include experimental results or at least a qualitative analysis on how label distribution characteristics, such as class imbalance, skewed distributions, or feature-label correlation may affect both attack susceptibility and SecDT’s defensive performance. Since many real-world datasets (e.g., medical, finance) are heavily imbalanced, addressing this would improve the robustness and generalizability of the proposed approach.
- Add a more detailed analysis, preferably with plots or tables, illustrating the trade-off between noise level (μ) and model performance (e.g., test accuracy, attack AUC). This would greatly assist practitioners in choosing appropriate noise settings based on their desired privacy-utility balance, and provide insight into SecDT’s tunability across use cases.

**Strengths And Weaknesses:**

Strengths:
- The use of fake labels via dimension expansion and randomized label-space mapping represents a novel and effective defense strategy. Notably, this approach operates without requiring modifications to the model architecture, making it adaptable and non-intrusive.
- By integrating directional protection, gradient norm flattening, and softmax-normalized perturbation, the method presents a comprehensive, layered defense. This multifaceted approach demonstrates a deep understanding of adversarial modeling and significantly strengthens privacy in split learning scenarios.
- The inclusion of a formal convergence proof adds theoretical robustness to the proposed method. It helps validate the algorithm’s reliability and justifies its application in practical, iterative training pipelines.

Weaknesses:
- The experiments adopt a fixed noise level (μ) across datasets, yet there is minimal discussion or guidance on how to tune this hyperparameter based on dataset characteristics or desired privacy-performance trade-offs.
- The paper does not explore how label distribution characteristics, such as class imbalance, label-feature correlation, or label skew may affect the attack vulnerability or the efficacy of SecDT. These factors could influence both privacy risk and model robustness.
- Some technical sections, particularly those describing dimension transformations and weighted label mappings, could benefit from enhanced clarity. Supplementing these with pseudocode, visual examples, or simplified walkthroughs would improve accessibility for a broader audience.
- There is limited analysis of scalability with respect to large K values. While increasing K reduces attack success, it may also introduce practical concerns such as increased computational complexity, memory usage, and potential convergence instability. A deeper empirical or theoretical examination of this trade-off would be valuable.

---

### Decision · Action_Editor_9yzf · 2025-06-04

**Recommendation:** Reject

**Additional Comments:**

This paper studied label leakage problems in split learning settings. For the initial submission, reviewers shared common concerns about the theoretical and empirical results to support the claims, yet the authors did not provide any responses. Therefore, I recommend rejection.

**Audience:**

Yes

**Audience Explanation:**

The topic is of sufficient interest to machine learning

**Claims And Evidence:**

No

**Claims Explanation:**

Some claims lack sufficient experimental and theoretical analysis